# Dually Responsive Nanoparticles for Drug Delivery Based on Quaternized Chitosan

**DOI:** 10.3390/ijms23137342

**Published:** 2022-07-01

**Authors:** Fenghui Qiao, Zhiqi Jiang, Wen Fang, Jingzhi Sun, Qiaoling Hu

**Affiliations:** 1MOE Key Laboratory of Macromolecular Synthesis and Functionalization, Department of Polymer Science and Engineering, Zhejiang University, Hangzhou 310027, China; qiaofh@zju.edu.cn (F.Q.); jiangzq@beforever.cc (Z.J.); 3140105178@zju.edu.cn (W.F.); sunjz@zju.edu.cn (J.S.); 2Center of Healthcare Materials, Shaoxing Institute, Zhejiang University, Shaoxing 312000, China

**Keywords:** chitosan, drug delivery, nanoparticle, dual response, dynamic bond

## Abstract

In this work, we report the fabrication and functional demonstration of a kind of dually responsive nanoparticles (NPs) as a potential drug delivery vector. The pH value, corresponding to the acidic microenvironment at the tumor site, and mannitol, to the extracellular trigger agent, were employed as the dually responsive factors. The function of dual responses was achieved by breaking the dynamic covalent bonds between phenylboronic acid (PBA) groups and diols at low pH value (pH 5.0) and/or under the administration of mannitol, which triggered the decomposition of the complex NPs and the concomitant release of anticancer drug of doxorubicin (DOX) loaded inside the NPs. The NPs were composed of modified chitosan (PQCS) with quaternary ammonium and PBA groups on the side chains, heparin (Hep), and poly(vinyl alcohol) (PVA), in which quaternary ammonium groups offer the positive charge for the cell-internalization of NPs, PBA groups serve for the formation of dynamic bonds in responding to pH change and mannitol addition, PVA furnishes the NPs with diol groups for the interaction with PBA groups and the formation of dynamic NPS, and Hep plays the roles of reducing the cytotoxicity of highly positively-charged chitosan and forming of complex NPs for DOX up-loading. A three-step fabrication process of drug-loaded NPs was described, and the characterization results were comprehensively demonstrated. The sustained drug release from the drug-loaded NPs displayed obvious pH and mannitol dependence. More specifically, the cumulative DOX release was increased more than 1.5-fold at pH 5.0 with 20 mg mL^−1^ mannitol. Furthermore, the nanoparticles were manifested with effective antitumor efficient and apparently enhanced cytotoxicity in response to the acidic pH value and/or mannitol.

## 1. Introduction

Chitosan (CS), as one of the most abundant natural macromolecules, has received considerable attention for its high biosafety, low immunogenicity, excellent biocompatibility, admirable biodegradability, and versatile biological activities [1,2,3,4,5,6,7,8,9]. In addition, CS is the only natural cationic polymer. This intrinsic structure allows CS and its derivatives to exhibit good water-solubility in wide pH range, strong muco-adhesive property, effective cellular uptake, passive targeting to tumor regions, and antibacterial activity. As a result, the cationic CS, or the quaternized chitosan (QCS), has recently found applications in the preparation of various materials, ranging from daily life (for instances, agriculture, water and waste, food and beverages treatment, cosmetics and toiletries) to advanced research (e.g., in biopharmaceutical and biomedical fields) [10,11,12]. One of the most promising applications is to utilize the intrinsic property of CS or QCS as matrixes in the fabrication of drug delivery nanocarriers [13,14,15,16,17,18]. Generally, drug delivery nanocarriers have recently garnered huge attention in many scientific branches [19]. Mihalache et al. synthesized a bupivacaine-loaded CS hydrogel as local anesthetics in clinical dentistry, which presented good bupivacaine inclusion and release capacity without cytotoxicity or side effects [20]. Dellali et al. fabricated a CS-based nanocapsule for drug delivery [21]. A detailed in vivo study conducted on experimental animals with three administration routes of intraperitoneal, subcutaneous, and oral demonstrated the biocompatibility of CS-based nanocapsules and their safe use in biomedical applications. Jin et al. fabricated mucoadhesive nanoparticles based on QCS and thiolated carboxymethyl CS, which demonstrated high hydrophilicity, enduring drug release, safety, and mucosal adhesion [22].

Despite its salient advantages, some key problems should be solved to develop QCS-based materials into practically applicable nanocarriers with advanced functions. One challenging problem is that the cationic QCS and the nanocarriers made from QCS have cytotoxicity and can be rapidly cleared by the phagocyte system [23]. Another challenge is to formulate the stimuli-responsive drug-delivery nanocarriers showing tunable changes to endo-/exogenous physicochemical changes such as pH, redox species, temperature, and light triggers [24,25,26,27,28,29]. However, QCS itself has no such kind of stimuli-responsive characteristics. Consequently, it is of great significance to furnish QCS with controllable stimuli-responsive property. To tackle these problems, in this work, we propose an unprecedented chemical strategy to fabricate drug delivery vector using QCS-based nanocarriers as substrate. The basic design idea is illustrated in Figure 1, and the fabrication procedures, experimental results, and working mechanism are described as follows.

## 2. Results

### 2.1. Synthesis and Characterization of Quaternized Chitosan Modified with Phenylboronic Acid

As illustrated by step I in Figure 1 and Appendix A, through EDC/NHS coupling reaction between the amino groups in quaternized chitosan (QCS) and carboxyl groups in 4-carboxyphenylboronic acid (PBA), quaternized chitosan modified with phenylboronic acid (PQCS) could be derived. The synthetic procedures are according to the literature and described in Appendix A [30]. The ^1^H NMR spectrum of the derived PQCS shows the chemical shifts of the protons which are consistent with the expected chemical structure [31]. The chemical shift around δ 7.7–7.8 ppm is assigned to the proton’s resonance of phenyl rings in PBA units, and the resonance at δ 3.2 ppm originates from the methylene groups linked with quaternary ammonium units (Figure 2a and inset). The degree of substitution of quaternary ammonium units and PBA units on PQCS were calculated as 46.1 mol% and 21.2 mol%, respectively (Appendix A). The FTIR spectra (Appendix A) and XPS survey spectrum (Appendix A) also indicated successful synthesis of PQCS (Appendix A).

Alizarin red (AR) was used as an indicator to estimate the concentration of boronic acid in a previous study [32]. Since the catechol groups in AR can bind to phenylboronic acid groups via esterification, AR shows conspicuous fluorescence emission and color change with the existence of boronic acid. As shown in Figure 2b, the fluorescence intensities of AR/PQCS solutions at 579 nm dramatically increased upon increasing the concentrations of PQCS. The similar phenomenon was observed in fluorescence spectra of AR/PBA solutions (Appendix A). Based on fluorometry assay, the degree of substitution of phenylboronic acid of PQCS was calculated to be 21.8 mol% (Appendix A), which is nearly identical to the result obtained above. Besides, a visible color change from orange to yellow with increasing of the concentrations of PQCS was observed in Figure 2c. The substitution of PBA in PQCS was further verified.

### 2.2. Preparation and Characterization of the Drug-Loaded Complex Nanoparticles

As shown by step II in Figure 1, the primary nanoparticles (NPs) of heparin (Hep) and PQCS were prepared by simple mixing the two components in aqueous solution. The polyelectrolyte complexes were formed by electrostatic interaction between the negatively and positively charged polysaccharides of Hep and PQCS, respectively. By adjusting the relative amount of Hep and PQCS, the Hep/PQCS complexes can bear the proper number of positive charges, and cationic complexes can be derived. The anticancer drug doxorubicin (DOX) could be incorporated inside the complex NPs just by physically blending drug and Hep solutions during the formation of Hep/PQCS NPs.

In step III (see Figure 1), when the cationic complexes was mixed with poly(vinyl alcohol) (PVA) in an aqueous solution with pH 7.4, chemical crosslinking between PQCS and PVA occurred because of the formation of ester bonds between the vicinal diol units on PVA chain and boric acid groups on surface of Hep/PQCS complex [30]. According to the fabrication process, the NPs would have a core-shell structure with Hep/PQCS polyelectrolyte complexes and PVA as core and shell, respectively.

The Z-average sizes and zeta potentials of Hep/PQCS polyelectrolyte complexes measured by dynamic light scattering (DLS) technique are shown in Figure 3a. With adding PQCS in DOX-containing Hep solution, the zeta potential of polyelectrolyte complexes solutions increased from negative to positive and the sizes increased before the isoelectric point and then decreased. The polyelectrolyte complexes at a Hep/PQCS volume ratio of 1/8 had appropriate size (171.7 ± 1.90 nm) and zeta potential (20.4 ± 0.49 mV), and thus they were chosen in subsequent processes. Onto the Hep/PQCS polyelectrolyte complex core, PVA was attached as a shell. As shown in Figure 3b, the zeta potentials of solutions continuously decreased with adding PVA solutions, because the non-ionic PVA layer moved the shear plane away from the charged surface. The increasing sizes of NPs with increasing amounts of PVA also gave evidence of PVA layer’s constitution. The NPs with 0.05 mg mL^−1^ PVA (194.9 ± 2.31 nm, 14.5 ± 0.32 mV) that maintained good cationic character were chosen in a subsequent study. Next, we proceeded to assemble PVA and PQCS successively in a layer-by-layer manner. As shown in Figure 3c, the sizes of NPs increased with the covalent layer number and the zeta potentials displayed a regular polygonal line which determined by the composition of the outermost layer. As described above, the formation of PVA layer decreased the magnitude of positive charge on the surface of NPs, whereas the formation of PQCS layer increased the positive charge density. These results demonstrated the core-shell structure of DOX-loaded Hep/PQCS-PVA assemblies.

The shell-core structure was further examined by TEM measurements (Figure 4), and the high-resolution images clearly display two features of the fabricated DOX-loaded Hep/PQCS-PVA assemblies—one is the nano-size and the other is the core-shell morphology of the nanostructures. Considering that the NPs are potentially used in physiological environment, the dispersibility in different media should be investigated. As shown in Figure 4a,b, the DOX-loaded Hep/PQCS-PVA NPs can be dispersed in deionized water and DMEM because the cationic nature repels the NPs from each other. In addition, the NPs all display intact morphology, which means the NPs are stable in both media. The sizes of DOX-loaded Hep/PQCS-PVA NPs in deionized water shown in TEM pictures range from 146 to 178 nm, which is smaller than the sizes measured by DLS (211.1 ± 2.37 nm, refers to Figure 3d). The difference is due to the shrinkage of hydrophilic NPs during the drying process. Compared to the size in deionized water, the sizes of NPs in DMEM decrease (60–69 nm, Figure 4b).

### 2.3. Dually Responsive Drug Release In Vitro

The DOX-loaded Hep/PQCS-PVA NPs expressed acute responses to pH value and mannitol in DMEM, owing to the susceptible phenyl-boric-diol ester linkages between core and shell. As shown in Figure 5a, the diameters of NPs became around 4 nm after incubation in DMEM at pH 5.0 for 30 min, indicating that the NPs severely dissociated in a weak acid condition. With reduction of pH value, negatively charged phenylboronate ester was transformed to neutral form, which was more vulnerable to hydrolysis, and the increase in hydrophilicity of PQCS was also adverse to the aggregation of NPs. Compared to the neutral physiological pH, the environment in the cancer tissues was more acidic (around pH 6.5) because of the accumulation of lactates as byproducts of aerobic glycolysis, and the environment became even more acidic in endosomes (pH 5.0) [33,34], which was beneficial to responsive drug release of the DOX-loaded Hep/PQCS-PVA NPs in the tumor sites. Meanwhile, mannitol containing three cis-diol pairs could also efficiently cleave the phenylboronate-diol ester bonds of the NPs, as evidenced by the reduction sizes of NPs (around 23 nm) shown in Figure 5b. The high concentration and affinity of mannitol in esterification reaction with phenylboronic acid might be the predominant reasons for the disruption of the nanostructure. Generally, mannitol is an FDA-approved drug which can be applied with recommended high dose as an on-demand cleavage reagent in vivo.

The release profiles from DOX-loaded Hep/PQCS-PVA NPs were estimated by standard procedures (see Section 4). The DOX loading efficiency was calculated to be above 98.8%, while the DOX loading content was around 9.4% (Appendix A). As shown in Figure 5c, sustained DOX releases were observed with burst releases in initial 0.5 h and the DOX burst were 7.6%, 16.2%, 18.2%, and 24.1% for DOX-loaded Hep/PQCS-PVA NPs under different conditions, respectively. The release rates of DOX gradually decreased after 5 h, and the cumulative release of DOX was 19.0% after 96 h for Hep/PQCS-PVA NPs in PBS with pH 7.4 (Figure 5d). The release of DOX was accelerated by either decreasing the medium pH or adding mannitol, on account of NP dissociation. When adjusting the pH value of PBS to 5.0 and with administration of mannitol (20 mg mL^−1^), the cumulative release of DOX from NPs after 96 h was 43.3% and 34.5%, respectively. The cumulative DOX release notably attained 50% for DOX-loaded Hep/PQCS-PVA NPs at pH 5.0 and existence of mannitol. The dually responsive DOX-loaded Hep/PQCS-PVA NPs to pH value and mannitol could be further exploited as anticancer carrier with rapid drug release triggered by acidic tumor and intracellular microenvironments, or by systemic intravenous injection of mannitol.

### 2.4. Cell Viability

To explore the prospects of DOX-loaded Hep/PQCS-PVA NPs in drug delivery field, their biocompatibility was first evaluated by MTT assay. As shown in Figure 6c, no obvious growth inhibition effect was found after incubation with the blank NPs, even at 262.5 μg mL^−1^ for 3 days. The viabilities of cells were more than 87% at the 3rd day against all cell lines in the experiment.

MTT assay was also conducted to compare toxicities of the DOX-loaded Hep/PQCS-PVA NPs and free DOX against Hep G2 cells and HeLa cells. As shown in Figure 6a,b and Appendix A, significant proliferation inhibition following 1 day and 3 days incubation was observed, reflecting the efficient antitumor effect of NPs. Moreover, the IC_50_ of DOX-loaded Hep/PQCS-PVA NPs after 1 day against Hep G2 was 4.18 μg DOX equiv. mL^−1^. The free DOX (IC_50_ = 2.24 μg mL^−1^) was slightly more toxic than NPs, which was due to the controlled drug release of NP carriers in cytoplasm and nonspecific cellular uptake of free DOX. However, this effect lessened with incubating, and the toxic of NPs and free DOX trended to be the same after 3 days. Furthermore, the toxicities of NPs and free DOX against FIB cells were investigated through MTT assay (c.f., Appendix A). The DOX-loaded Hep/PQCS-PVA NPs displayed more efficient proliferation inhibition toward Hep G2 cells and HeLa cells than FIB cells.

As described above, the DOX-loaded Hep/PQCS-PVA NPs were dually responsive and could break up and release DOX under stimulation of acidic environment or mannitol. The NPs or free DOX was sequentially triggered by acidic pH and/or 20 mg mL^−1^ mannitol when they were incubated with Hep G2 cells at concentration of DOX of 1.5 μg mL^−1^. There were inappreciable changes in the toxicity profile of free DOX in stimulation (Appendix A). On the contrary, the NPs showed significantly enhanced cancer cell viability inhibition at pH 5.0 and with administration of mannitol (Figure 6d). Noteworthily, the cell viability inhibition of NPs in stimulation was slightly stronger than free DOX after 1 day of incubating, possibly because the cationic NPs easily aggregated around the negatively charged cancer cells and released drugs in response to affect regions with high DOX concentration in a short time. The cell viability inhibitions of NPs and free DOX became comparable after incubating for 3 days. By reason of the foregoing, Hep/PQCS-PVA NPs demonstrated the capacity to deliver anticancer drugs efficiently and release drugs on-demand with dually responsive character.

## 3. Discussion

PBA is a boron-containing organic compound with diol binding property, which is fast and reversible in stimuli of pH or chemical molecules with diol groups. When PBA-diol compounds are at a lower pH, for instance at the tumor site which with lower pH value than that of normal tissue [35], there is a shift in the equilibrium from negatively charged boronate esters to the hydrolyzed neutral form of boronic acids [36]. It is noteworthy that the shift may change the structures and properties (like solubility and fluorescent property) of PBA-diol compounds. Therefore, PBA-based polymers were widely used as stimuli-responsive devices for drug delivery based on esterification with diols such as curcumin [37], polyvinyl alcohol (PVA) [38,39], and so on. The administration of competing diols as exogenous stimuli was also a strategy to trigger drug release of PBA-diol nanocarriers [32,40]. PBA-functionalized quaternized chitosan (PQCS) was derived by grafting PBA groups onto the polymer chain of QCS successfully. According to the characterization data (Figure 1 and Appendix A), the target structure of PQCS was confirmed. This was a crucial step because it determined the formation of dynamic covalent bonds between the Hep/PQCS complexes and PVA and finally determined the dual responses to pH value and mannitol addition.

The Hep/PQCS-PVA nanoparticles were prepared by physical and chemical crosslinking among heparin, PQCS and PVA. Hep is a negatively charged polysaccharide with strong anticoagulant activity and anticancer activity [41,42]. Positively charged QCS is often used in delivery and recovery of Hep in bio-pharmaceutical field [43]. Herein, the positively charged PQCS interacted with heparin to form electrostatic complex nanoparticles (Hep/PQCS) firstly, and then the PVA was bound with Hep/PQCS with the method of PBA-diol esterification. The covalent ester bonds formed between diol and PBA groups could break down in response to pH lowering and mannitol addition, thus the Hep/PQCS-PVA NPs are expected to possess dually responsive characteristics. The preparation of Hep/PQCS and Hep/PQCS-PVA NPs were characterized with multiple spectroscopic methods and micro-morphological observation techniques, including TEM, Z-average size, and zeta potential analyses. The model anticancer drug DOX was loaded into the NPs in the 1st step of Hep-PQCS complex formation. By controlling the concentration of Hep and PVA, NPs with core-shell structure, appropriate particle size (194.9 ± 2.31 nm), and positive zeta potential (14.5 ± 0.32 mV), to the benefit of uptake by cancer cells [44,45], were obtained (Figure 3). The NPs had superior dispersing in physiological conditions like DMEM than in deionized water because many dissociative charged ions, proteins, and saccharides in DMEM could interact with PQCS. The NPs with excellent dispersibility are potentials for practical applications like drug delivery.

Sustained drug release from the DOX-loaded Hep/PQCS-PVA NPs was observed which was achievable in general CS-based vehicles [22,46]. As shown in Figure 5a,b, the NPs in stimuli of pH = 5 or 20 mg mL^−1^ mannitol severely dissociated due to the dynamic nature of the borate bond between PQCS and PVA. Consequently, the cumulative DOX release was accelerated by either decreasing the pH value of media or adding mannitol (Figure 5c,d). The significantly enhanced cytotoxicity of DOX-loaded Hep/PQCS-PVA NPs against Hep G2 cells at pH 5.0 and with administration of mannitol was also observed (Figure 6d). This dually responsive NPs can react in a dynamic method when recognizing not only the acidic microenvironments in tumor sites but also exogenous stimulus of mannitol, whereas many studies of stimuli-responsive drug delivery based on CS focused on endogenous stimuli like pH value, redox condition, and enzyme [47,48,49].

It is noticed that the composing components of the nano-carrier, including heparin, quaternized chitosan, PVA, and even the trigger agent of mannitol, are biocompatible and have good biosafety [20,21,38,41]. The key component of PQCS is easily available and biocompatible as a CS derivative [30]. As evaluated by MTT assay, the blank Hep/PQCS-PVA NPs were identified with good cyto-compatibility, as expected (Figure 6c). The cytotoxicity of the DOX-loaded Hep/PQCS-PVA NPs and free DOX toward Hep G2, HeLa, and FIB cells was also evaluated (Figure 6 and Appendix A). The efficient antitumor effect of DOX-loaded NPs was confirmed and the slightly lower cytotoxicity of NPs than free DOX was consistent with the controlled drug release profiles [50]. Besides, the DOX-loaded Hep/PQCS-PVA caused less cell viabilities in cancer cells (Hep G2 and HeLa cells) than FIB cells, which could be ascribed to the acidic extracellular pH of cancerous cells [33,34], negatively charged cancerous cells membranes [44], and the targeting effect of PBA groups [51].

## 4. Materials and Methods

### 4.1. Materials and Reagents

Chitosan (CS) was purchased from the Qingdao Haihui Bioengineering Co., Ltd. (Qingdao, China) with 81.85% degree of de-acetylation and average viscosity molecular weight (Mη) of 3.04 × 10^4^ Da. PBA, 1-(3-Dimethylaminopropyl)-3-ethyl-carbodiimide hydrochloride (EDC), N-hydroxysuccinimide (NHS), PVA (with 98.0–99.0% degree of alcoholysis and viscosity of 3.2–3.8 mPa s), heparin sodium salt (185 USP units mg^−1^), and doxorubicin hydrochloride were purchased from Shanghai Aladdin Biochemical Technology Co., Ltd. (Shanghai, China). The other chemicals and reagents were all purchased from Aldrich (St. Louis, MI, USA) or Sinopharm Chemical Reagent Co., Ltd. (Shanghai, China) and were used without further purification. Dulbecco’s modified eagle medium (DMEM) containing 10% fetal bovine serum and antibiotics (100 units mL^−1^ penicillin and 100 μg mL^−1^ streptomycin) were supplied by Geno Bio-pharmaceutical Technology Co., Ltd. (Hangzhou, China). Deionized water was provided by Zhejiang University.

### 4.2. Synthesis of PQCS

PQCS was prepared in procedures similar to those previously reported [22,30]. 2,3-Epoxypropyltrimethylammonium chloride (GTA, 0.8 mL) diluted with deionized water (1 mL) was added into a CS acetic acid aqueous solution (10 mL, 50 mg mL^−1^). After stirring at 70 °C for 8 h, the product was dialyzed with deionized water and then lyophilized. QCS (0.1 g) was dissolved with 100 mL deionized water in a 250 mL beaker. Add PBA (0.1 g), NHS (1 g), and EDC (2 g) into the beaker and stir the mixture for 3 h at room temperature. The pH of the solution was adjusted between 5.5 and 6.0 with 1 M NaOH. The product was dialyzed with deionized water and finally lyophilized. The structure and the degree of substitutions (D.S.) of the PQCS product were determined by ^1^H NMR spectroscopy with a Bruker Avance 500 MHz spectrometer, FTIR spectroscopy with a Bruker Vector 22 spectrometer and X-ray photoelectron spectroscopy (XPS) on a Thermo Scientific ESCALAB 250 Xi instrument with Al Kα (1486.6 eV) source. All the binding energies were referenced to the C1s peak at 284.8 eV. The PQCS PBS solutions were mixed with alizarin red (AR) indicator PBS solution (0.04 mg mL^−1^). Fluorescence spectra were recorded on a Perkin-Elmer LS 55 spectrofluorometer with xenon discharge lamp excitation.

### 4.3. Preparation of Hep/PQCS-PVA and Drug-Loaded Hep/PQCS-PVA NPs

A calculated amount of heparin aqueous solution (1 mg mL^−1^) was added into preset PQCS (1 mg mL^−1^) aqueous solution with magnetic stirring for 30 min to form electrostatic composite NPs. Then, the same volume of PVA aqueous solution was added into the mixture solution dropwise subsequently. Control the pH of the mixture solution at 7.4 and react for 30 min. DLS particle size (Brookhaven, BI-90Plus, Holtsville, NY, USA) and Zeta potential (DelsaTM, Beckman Coulter, Brea, CA, USA) were examined to characterize the generated NPs. Furthermore, TEM measurement was conducted on a JEM 1200EX transmission electron microscope.

For drug-loaded Hep/PQCS-PVA NPs, DOX was selected as a model drug. DOX-loaded Hep/PQCS-PVA NPs were prepared by mixing a stoichiometric amount of DOX DMSO solution (1 mg mL^−1^) with heparin solution firstly, and then following the same procedures as preparing the unloaded NPs. The unload DOX was removed by centrifuging in ultrafiltration centrifuge tube (3000 Da MWCO) at 4000 rpm for 1 h. The DOX-loaded NPs on filters were recovered by PBS and the filtrate was applied to measure the concentration of unloaded DOX on a Tecan Infinite M200. The pH values of DOX loaded Hep/PQCS-PVA NP solutions were fine-tuned by HCl solution (0.1 M) and NaOH solution (0.1 M).

### 4.4. Drug Release Estimation

1 mL 0.1 mg mL^−1^ DOX-loaded NPs solution was injected into a dialysis bag with a 3500 Da MWCO. The bag was dialyzed against 10 mL PBS at different pH levels (pH 7.4 and pH 5.0) or in the presence of concentrations of mannitol (0 and 20 mg mL^−1^). The system was incubated in a thermostatic oscillation incubator at speed of 100 rpm and temperature of 37 °C. The concentration of DOX releasing into the dialysis medium at various time points was measured on a Tecan Infinite M200. The fluorescence intensity at 563 nm was recorded with exciting at 475 nm. Values were reported as the means for each duplicate sample.

### 4.5. Cell Toxicity Evaluation

The cytotoxicity of empty and DOX loaded Hep/PQCS-PVA NPs was estimated by MTT assay. Human cervical carcinoma (HeLa) cells (4000 per well), human liver carcinoma (HepG2) cells (4000 per well), and human fibroblast (FIB) cells (8000 per well) were cultured with DMEM medium (0.2 mL per well) in 96-well plates overnight at 37 °C, 5% CO_2_, respectively. Then, the culture medium was replaced by the NPs DMEM solutions with different concentrations at pH 7.4 or 5.0, in the absence or in the presence of 20 mg mL^−1^ mannitol. The cells were continually cultured for 1 day or 3 days. We washed the cells with PBS buffer for three times and added 0.2 mL MTT DMEM solution (0.5 mg mL^−1^) to each well. The cells were cultured for additional 4 h. The culture medium was then replaced by DMSO (0.2 mL) to dissolve the purple formazan crystals. The absorbance at 570 nm was measured to calculate the viability of cells on a Tecan Infinite M200. Triplicate experiments were performed each time, and cells dealt without DOX or NPs were cultured as a blank control group.

## 5. Conclusions

In this work, we synthesized a new CS derivative, PQCS, and developed a facile method for preparing the Hep/PQCS-PVA nanoparticles through electrostatic interactions between Hep and PQCS and borate cross-linkages between PQCS and PVA. By tunning the concentrations of Hep and PVA in the mixture, we obtained nanoparticles with core-shell structure, appropriate particle size (194.9 ± 2.31 nm), and proper zeta potential (14.5 ± 0.32 mV) for dually sensitive break-up and favorable cellular uptake. DOX was employed as an anticancer model drug and loaded into the nanoparticles. Sustained drug release from the DOX-loaded Hep/PQCS-PVA nanoparticles was observed, and the cumulative DOX release could be accelerated by either decreasing the pH value of media or adding mannitol. Next, the cytotoxicity of the nanoparticles toward Hep G2, HeLa, and FIB cells was evaluated by MTT assay, and the blank Hep/PQCS-PVA nanoparticle was identified with good cytocompatibility. The nanoparticles showed high anticancer efficiency with 4.18 μg DOX equiv. mL^−1^ of IC50 against Hep G2 cells, which could even be considerably raised in response to acidic pH value and/or mannitol. However, the results are still in the primary stage and model animal tests are awaited in the future. As a concept-proof investigation, the results suggest that chemical design for the nano-carrier is workable and the derived Hep/PQCS-PVA NP is a promising candidate for dually stimuli-responsive drug delivery in cancer therapy with activated drug release in the acidic tumor environments and the intravenous administration of mannitol.

## Figures and Tables

**Figure 1 ijms-23-07342-f001:**
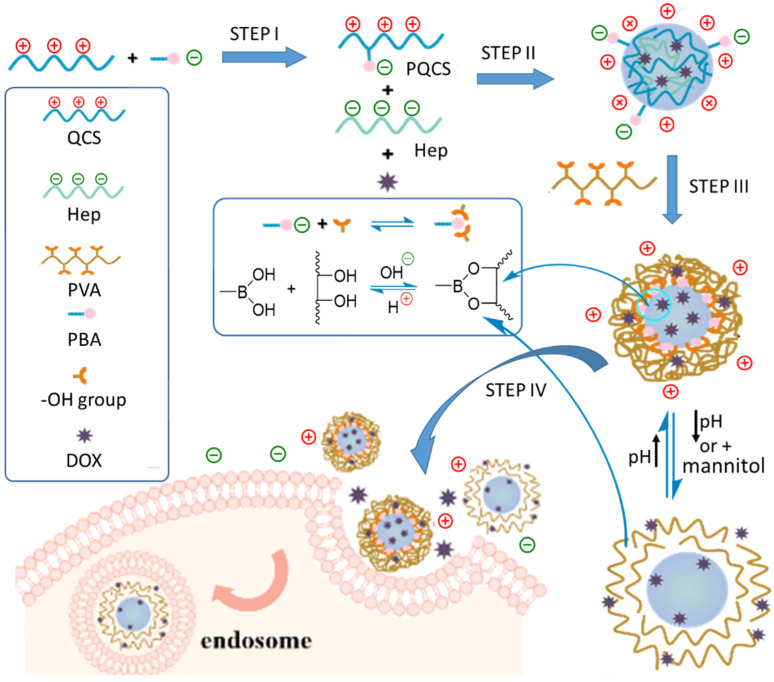
A schematic illustration of the procedures for the preparation of DOX-loaded nanoparticles (NPs), the mechanism of the dually responsive drug release, and cellular uptake of NPs. QCS = quaternized chitosan; PQCS = PBA-modified QCS; PBA = phenylboronic acid; Hep = heparin; PVA = poly(vinyl alcohol); DOX = doxorubicin.

**Figure 2 ijms-23-07342-f002:**
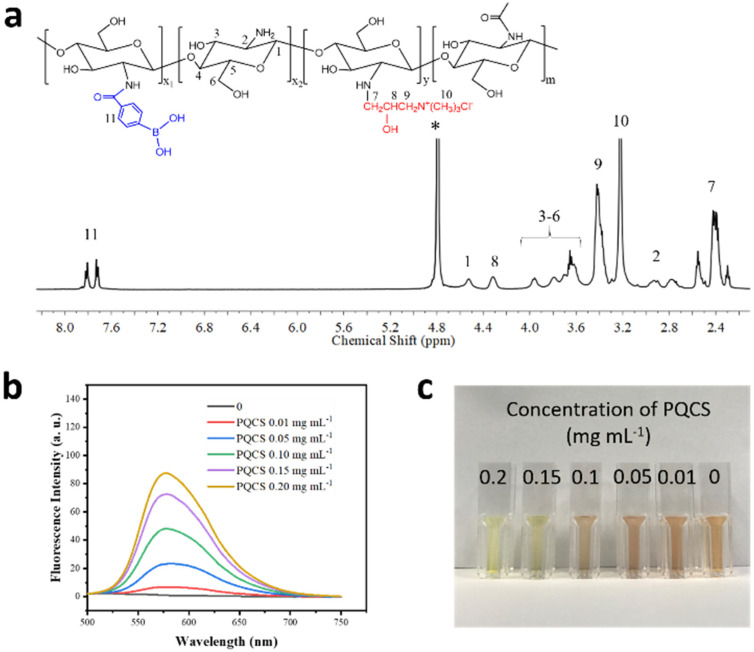
(**a**) ^1^H NMR spectrum for PQCS in D_2_O, the inset shows the chemical structure of PQCS. The solvent peak are marked with asterisk (*). (**b**) Fluorescence spectra of AR (0.034 mg mL^−1^)/PQCS PBS solutions, excitation: 468 nm; (**c**) photograph of AR (0.034 mg mL^−1^)/PQCS aqueous solutions.

**Figure 3 ijms-23-07342-f003:**
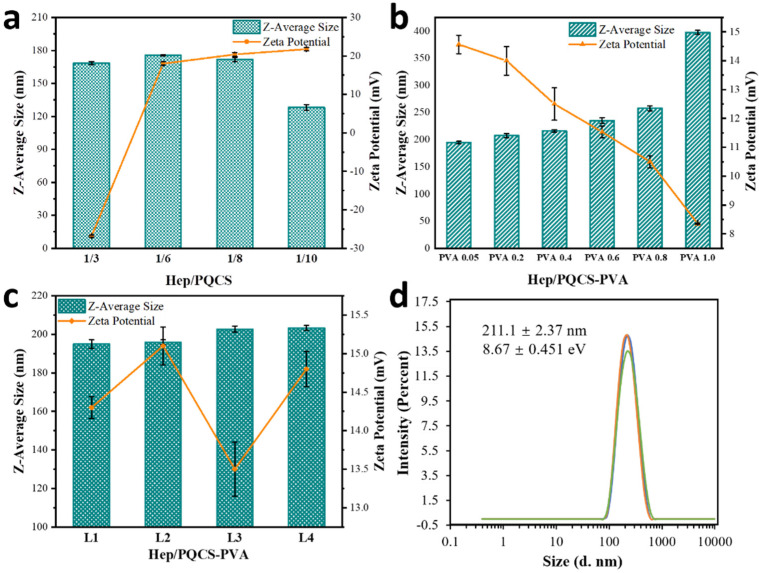
Characterization of DOX-loaded Hep/PQCS-PVA NPs. (**a**) Z-average sizes and zeta potentials of Hep/PQCS aqueous solutions, (**b**) DOX-loaded Hep/PQCS-PVA aqueous solutions, and (**c**) DOX-loaded Hep/PQCS-PVA multilayer NPs in aqueous solutions measured by DLS; (**d**) The particle size of DOX-loaded Hep/PQCS-PVA NPs aqueous solutions measured by dynamic light scattering (DLS) for three times. The concentration of each solution: 0.1 mg mL^−1^.

**Figure 4 ijms-23-07342-f004:**
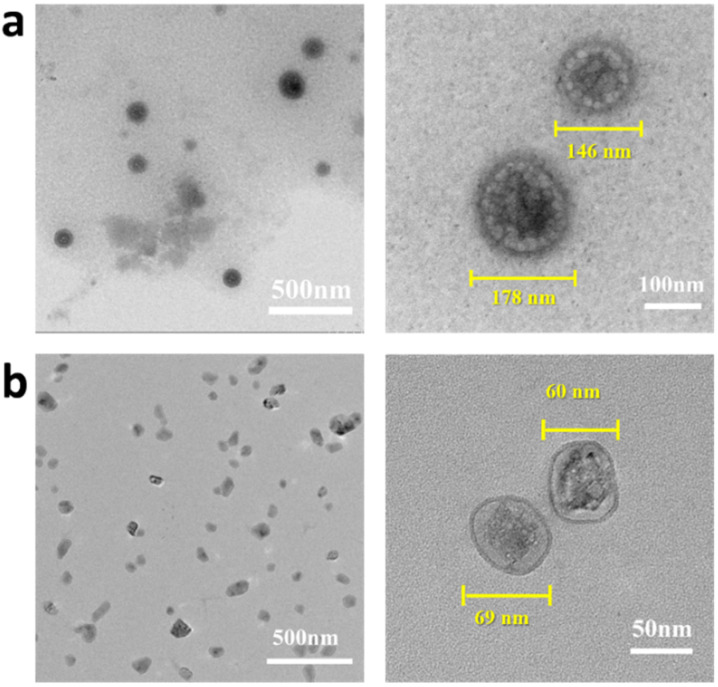
TEM images of the DOX-loaded Hep/PQCS-PVA NPs, the samples came from (**a**) deionized water and (**b**) DMEM. The images on the right are taken in high resolution mode.

**Figure 5 ijms-23-07342-f005:**
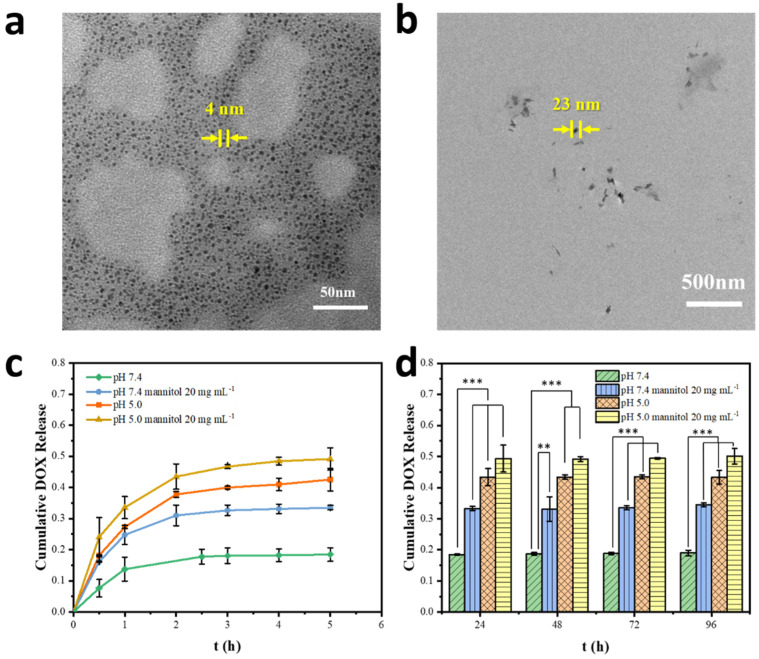
Dually responsive DOX release in vitro. (**a**,**b**) TEM images of DOX-loaded Hep/PQCS-PVA NPs in DMEM solutions at pH 5.0 and in the presence of mannitol (20 mg mL^−1^). (**c**,**d**) Cumulative release of DOX from the DOX-loaded Hep/PQCS-PVA NPs with or without treatment with 20 mg mL^−1^ mannitol at pH 7.4 or pH 5.0. **: *p* < 0.01, ***: *p* < 0.001.

**Figure 6 ijms-23-07342-f006:**
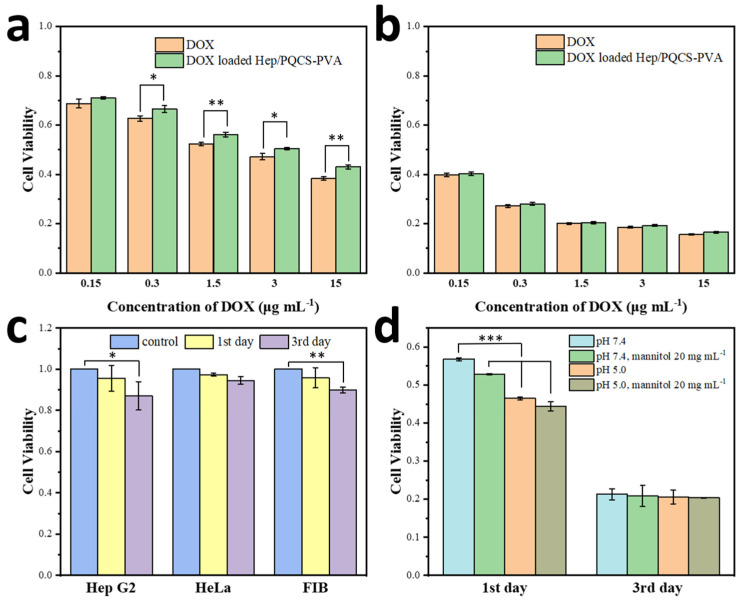
Cell viability measured by MTT assay. Cell viability of DOX-loaded Hep/PQCS-PVA NPs and free DOX toward Hep G2 cells at (**a**) 1st day and (**b**) 3rd day. (**c**) Cell viability of empty Hep/PQCS-PVA NPs toward Hep G2 cells, HeLa cells, and FIB cells. (**d**) Cell viability of DOX-loaded Hep/PQCS-PVA NPs toward Hep G2 cells with or without treatment with 20 mg mL^−1^ mannitol at pH 7.4 or pH 5.0. *: *p* < 0.05, **: *p* < 0.01, ***: *p* < 0.001.

## Data Availability

Not applicable.

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
