# Peer review of "Dually Responsive Nanoparticles for Drug Delivery Based on Quaternized Chitosan"

_ijms, 2022, doi:10.3390/ijms23137342_

Round 1

Reviewer 1 Report

Very interesting and original article. The tasks were solved with the help of modern methods.

Some Wishes:

Figure 3 has a very low resolution. The legend is not readable.

Figure 6. I would recommend just using different colors, no line.

It is necessary to draw a conclusion from the results of the research. 5. Conclusions

The discussion section should be written in more detail and in accordance with generally accepted rules. In 2022, 35 articles on the effects of chitosan have already been published. The discussion should include contemporary references to other authors' work that may be consistent with and support the authors' conclusions.

Reviewer 2 Report

The paper submitted by Qiao et al. deals with the multistep preparation and characterization of dually responsive nanoparticles based on quaternized chitosan and PVA. Moreover, these NPs were loaded with a model drug, doxorubicin. 

The manuscript is clear, well written and the conclusions are supported by the results. However, some corrections are needed before its publication:

1. the introduction section might be completed with other new references concerning the use of CS for the preparation of drug delivery systems. Some suggestions are: https://doi.org/10.1002/pi.6052; https://doi.org/10.3390/polym14091811

2. in the present form, the discussion section is only an extended conclusion section. in this section, the authors must discuss their results by comparing them to other literature data.

3. the conclusion section is missing. 

Author Response

Please find our responses in the attached file. thanks!

Reviewer 3 Report

In this work, fabrication and applicability of NPs being a drug delivery vector is presented.

Such topic to bring novel drug delivery compounds in the way to effective ensure transportation of pharmaceutical to the site-of-action in the organism without or minimal loss is of huge interest, now.

These carriers based on nanoparticles ensure sufficient drug application in treatment of various diseases.

Here, different analytical approaches were used to study features of the developed drug delivering vector, to validate its applicability.

REMARKS

1. I miss to add a general statement about the importance of drug delivery systems. Please, follow the down below upgrade with particular work to cite, in the first paragraph of introduction.

„One of the most promising applications is to utilize the intrinsic property of QCS as a matrix in the fabrication of drug delivery nanocarriers [13-18]. Generally, drug delivery nanocarriers have recently taken a huge attention in many scientific branches [10.1038/s41598-021-99678-y].

Despite of its salient advantages, some key problems should be solved to develop QCS based materials into practically applicable nanocarriers with advanced functions.”

2. I have not found a conclusion points dealing with the main outcomes and also the future aims of the authors in this scope should be given to the readers.

Author Response

Please find our responses in the attached file. Thanks!

Round 2

Reviewer 1 Report

The article has been significantly improved. However, there is one more note. Statistical methods for comparing results should be applied. Reliability and p-values should be presented in figures.

Author Response

Thanks a lot for your review and appreciate comments on our work. We have revised the Figure 5 d and Figure 6 a, c and d, in which the statistically significant differences (p-values) are indicated with asterisks.

Reviewer 3 Report

The remarks have Been adressed.

Author Response

Thank you for your review and appreciate comments on our work.

Round 3

Reviewer 1 Report

Thanks to the authors for the work done and understanding of my comments. All comments were taken into account and the article is worthy of publication in the journal.